Corrected: Publisher correction

# Online photochemical derivatization enables comprehensive mass spectrometric analysis of unsaturated phospholipid isomers

Wenpeng Zhang [1,2], Donghui Zhang[1], Qinhua Chen[3], Junhan Wu[1], Zheng Ouyang [1,2] & Yu Xia[1,2]

Mass spectrometry-based lipidomics is the primary tool for the structural analysis of lipids but the effective localization of carbon–carbon double bonds (C=C) in unsaturated lipids to distinguish C=C location isomers remains challenging. Here, we develop a large-scale lipid analysis platform by coupling online C=C derivatization through the Paternò-Büchi reaction with liquid chromatography-tandem mass spectrometry. This provides rich information on lipid C=C location isomers, revealing C=C locations for more than 200 unsaturated glycerophospholipids in bovine liver among which we identify 55 groups of C=C location isomers. By analyzing tissue samples of patients with breast cancer and type 2 diabetes plasma samples, we find that the ratios of C=C isomers are much less affected by interpersonal variations than their individual abundances, suggesting that isomer ratios may be used for the discovery of lipid biomarkers.

[1] MOE Key Laboratory of Bioorganic Phosphorus Chemistry & Chemical Biology, Department of Chemistry and State Key Laboratory of Precision Measurement Technology and Instruments, Department of Precision Instrument, Tsinghua University, Beijing 100084, China. [2] Department of Chemistry, Purdue University, West Lafayette, Indiana 47907, USA. [3] Affiliated Dongfeng Hospital, Hubei University of Medicine, Shiyan, Hubei Province 442000, China. Correspondence and requests for materials should be addressed to Z.O. (email: ouyang@tsinghua.edu.cn) or to Y.X. (email: xiayu@tsinghua.edu.cn)

ipids are an essential class of biomolecules that are heavily recruited in cell structuring, signaling, organelle compartmentalization, and energy production in different biological systems[1]. The cellular lipidome is highly complex, characterized by the presence of thousands of molecular species with concentrations spanning three to five orders of magnitude[2]. The exact lipid composition, however, is closely regulated for maintaining appropriate cell functions regardless of differences in feed nutrients[3]. On the other hand, disturbed lipid homeostasis is often linked to the occurrence and development of a pathological state, as commonly found for cancer[4], insulin-resistant diabetes[5], and cardiovascular diseases[6]. Therefore, mapping the complete molecular composition of a lipidome is considered as an important goal of lipidomics in order to further understand pathways and mechanisms behind lipid homeostasis[7]. Mass spectrometry (MS) has become the enabling tool for lipidomics due to the capability of identifying and quantifying lipids in complex mixtures at high sensitivity and throughput[1]. Lipid profiles generated by MS are increasingly utilized in metabolic flux analysis, tissue imaging, and disease biomarker discovery[8,9].

Even though lipids, using glycerophospholipids as an example, are assembled from simple chemical building blocks, a plethora of structural possibilities exist when considering different combinations of acyl/alkyl chains, their linkage positions on the glycerol backbone (*sn*-position), and the occurrence of carbon–carbon double bonds (C=C) and stereo-centers. With the development of modern mass spectrometry and data analysis tools, large-scale lipid profiling can be routinely practiced at fatty acyl/alkyl compositional level[1,10], but not at the C=C location level. This limitation stems from employing low-energy collision-induced dissociation (CID) as the tandem mass spectrometry (MS/MS) technique in conventional lipidomics analysis workflows, which fails in providing the information for locating C=C bonds. Considering that the biophysical properties of unsaturated lipids are closely modulated by the number, locations, and configurations of C=Cs in acyl/alkyl chains[11], research in lipidomics is hampered by the limited knowledge about the exact structures of unsaturated lipids. Several MS strategies have attempted to address this challenge. These include C=C specific chemical derivatization or cleavage prior to MS[12–14], coupling high resolving power ion mobility separation with MS[15], and alternative gas-phase ion activation methods either producing fragments specific to a C=C[16–19] or inducing extensive fragmentation along the acyl chain[20]. While the newly developed gas-phase fragmentation methods are promising in terms of indeendent identification of unknown structures from complex mixtures, their capability in global identification of unsaturated lipids remains to be demonstrated.

In earlier studies, we have developed a strategy that combines solution C=C specific derivatization (the Paternò–Büchi (PB) reaction) with conventional MS/MS methods via low-energy CID, termed as PB–MS/MS. The capabilities of PB–MS/MS for localization of C=C in unsaturated lipids (free fatty acids (FAs), cholesterol esters, and glycerol phospholipids (GPs))[21–24] and quantitation of lipid C=C location isomers have been demonstrated using the shotgun lipid analysis approach, where crude lipid extract is directly subjected for MS analysis without a prior separation. Considering that liquid chromatography (LC–)MS is one of the most commonly used analytical platforms for large-scale lipidomic studies, merging the PB reaction onto the LC–MS/MS platform would provide a key solution to unsaturated lipid analysis at C=C location level.

Herein, we report the development of an online-coupled LC–PB–MS/MS platform and corresponding analytical workflows, aiming to achieve large-scale lipid analysis at C=C location level in a sensitive and high-throughput fashion. One distinct capability of the LC–PB–MS/MS platform is to monitor the relative compositional changes of lipid C=C location isomers at high precision (relative standard deviation (RSD) ~15%), without requiring the use of internal standards. This function enables the discovery of significant changes in relative isomeric ratios for a series of lipid C=C location isomers from human breast cancer tissue and type 2 diabetes (T2D) plasma samples. This type of information would not be readily accessible using conventional lipid profiling methods. Our findings suggest that the LC–PB–MS/MS can serve as a screening tool for the discovery of potential lipid biomarkers for clinical and point-of-care analysis.

## Results

**The LC–PB–MS/MS system**. We choose to implement the PB reaction post-LC separation while immediately before electrospray ionization (ESI), because this configuration preserves the retention time (RT) of each lipid subclass, regardless of possible changes in the PB reaction solvent system (Fig. 1a). This aspect facilitates the development of data dependent LC–PB–MS/MS analysis workflow as described later. Regarding the LC system, hydrophilic interaction chromatography (HILIC) was employed for lipid class separation and the PB reagent, acetone, was directly employed in the mobile phase. Good separations of different subclasses of GPs were obtained by mixing 40–50% (v%) acetone with ACN/$H_2O$ (Supplementary Fig. 1). This gradient elution is adequate for separating most phospholipids in biological samples, including PE, PC, PG, PA, PI, PS, and SM[25].

**The flow microreactor**. Ultraviolet (UV) transparent fluorinated ethylene propylene (FEP) tubing (0.03-in. i.d., 1/16 o.d.) was used to as the flow path in the flow microreactor (Fig. 1a and details in Supplementary Fig. 2)[26]. Briefly, the FEP tubing (~30 cm in length) was concentrically coiled at a diameter of 2.5 cm and supported on a stainless-steel rack; a low-pressure mercury lamp was installed at the center of the rack. This configuration was of a relatively compact size (reactor dimension of 30 mm × 30 mm × 80 mm) and it allowed an even exposure of the reaction system to UV irradiation over the entire flow path. The performance of the FEP flow microreactor for the PB reaction was evaluated using model lipid PC 16:0/18:1(9Z) under similar solvent compositions as used in HILIC separation. We found that increasing acetone content above 40% (v%) provided only marginal improvement of the PB reaction yield, but at a cost of decreased efficiency of HILIC separation (Supplementary Fig. 3). In addition, side reaction products derived from Norrish type I cleavage of acetone became more abundant, leading to increased chemical interference. Under the optimized solvent condition (acetone/ACN/$NH_4Ac$ (10 mM, aqueous) (40/40/20, v/v/v)), the PB reactions of a variety of GP standards, belonging to different subclasses and containing different degrees of unsaturation, could all reach a steady state in 40 s with moderate yields (20–30%). It is worth noting that the PB products resulting from single acetone addition, the preferred form for C=C location determination by PB–MS/MS[22], constituted as the primary reaction products under the optimized reaction condition (>90%, Supplementary Fig. 3f). These results demonstrate that the PB reaction could be reliably implemented online with HILIC–LC–MS.

**Workflow for structural identification of GP at C=C level**. Figure 1b shows an overview of the analysis workflow for structural identification of GP at C=C location level on a 4500 QTRAP mass spectrometer. The data collection process consisted of two LC–MS/MS runs and one data dependent LC–PB–MS/MS run. A home-built data analysis software, Lipid Omega Analyzer (LOA), was developed to guide data collection and perform

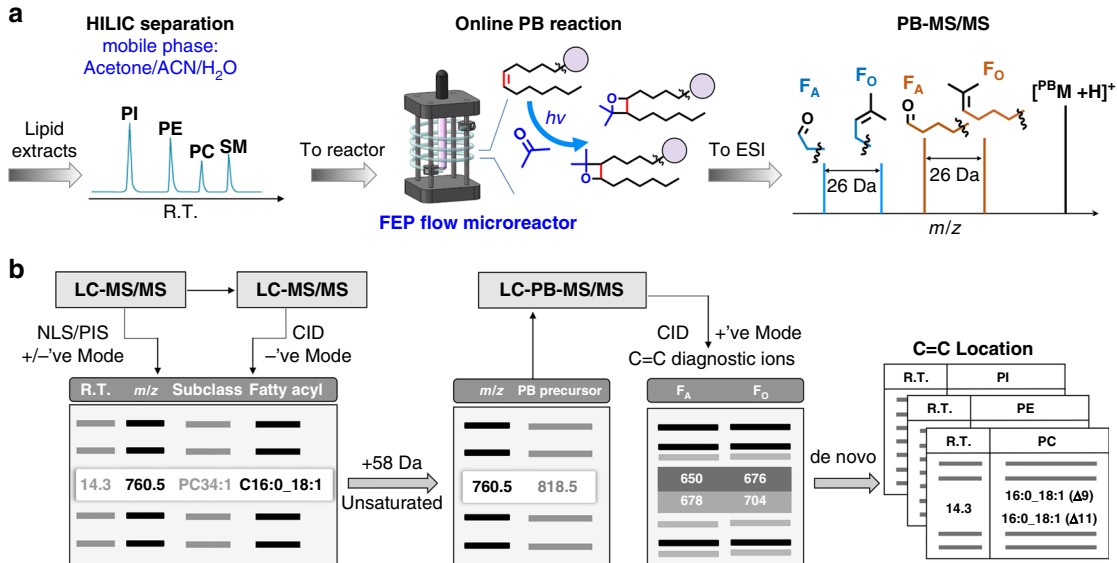

**Fig. 1** Schematic of the LC–PB–MS/MS system for lipid identification at C=C location level. **a** The key components include lipid class separation on a HILIC column employing acetone/ACN/NH$_4$Ac (10 mM) aqueous solution as mobile phase, a flow microreactor made from FEP tubing and a low-pressure mercury lamp (emission at 254 nm wavelength) for online coupling with LC and ESI–MS, and formation of C=C diagnostic ions via CID of the PB products of unsaturated GPs (PB-MS/MS). **b** The analysis workflow consists of two conventional LC–MS/MS runs for lipid identification at fatty acyl level and data dependent LC–PB–MS/MS for lipid identification at C=C location level. Data analysis was conducted using a home-developed program, Lipid Omega Analyzer

structural identification at different levels (Supplementary Figs. 4 and 5). The first LC–MS/MS run employed linked scans to identify lipids at the lipid subclass level, based on the identity of the headgroup, the number of carbons in side chains, and the degrees of unsaturation. LOA read the data and extracted a list of lipids with information of *m/z* value, monoisotopic molecular mass, headgroup, and RT. This list was used for generating precursor ion information ([M-H]$^-$ for PEs, PGs and PIs, and [M+CH$_3$COO$^-$] for PCs) for data dependent analysis in subsequent negative ion mode LC–MS/MS experiments. LOA analyzed the data and provided identification of GPs at the fatty acyl level (Supplementary Note 1). A sublist consisting only unsaturated GPs was extracted by LOA to generate a precursor ion list corresponding to their PB reaction products ($^{PB}$M), using the signature 58-unit increase in *m/z* due to single acetone addition. This list was then imported into the Analyst software for subsequent LC–PB–MS/MS via beam-type CID. In a typical PB–MS/MS spectrum, a pair of C=C diagnostic fragment ions, denoted as F$_A$ (consisting of an aldehyde) and F$_O$ (containing an olefin functional group), should be observed at similar ion intensities for each C=C (structure shown in Fig. 1a). Based on the detection of these C=C diagnostic ions, de novo analysis was performed by LOA and the locations of C=C in unsaturated GPs were assigned. If C=C location isomers were detected, LOA also conducted relative quantitation of the isomers (Supplementary Note 1).

**Analysis of GP from bovine liver polar extract**. The LC–PB–MS/MS system was tested with commercially available polar lipid extract from bovine liver. As shown in Fig. 2a, different subclasses of GP, including PI, PE, PC, and SM, were well separated in a 20 min-run through a HILIC column (0.1 μg lipid injection per run). Using the procedure for structural identification of PE and PC as an example, lipid subclass information was obtained from 141 Da neutral loss scan (NLS) for PEs and 184 *m/z* precursor ion scan (PIS) for PCs, in positive-ion mode (Fig. 2b, c). The fatty acyl composition for each lipid species was obtained from LC–MS/MS in negative ion mode. For instance, the peak at

*m/z* 718.6 in 141 Da NLS (monoisotopic molecular mass: 717.6 Da, RT: 5.2 min) was identified as PE 16:0_18:1, based on the detection of fatty acyl anions at *m/z* 255 (C16:0) and *m/z* 281 (C18:1) (Supplementary Fig. 6). The *sn*-positions of fatty acyls could not be confidently identified and thus are not reported herein. In subsequent LC–PB–MS/MS analysis, the protonated PB reaction product of PE 16:0_18:1 (*m/z* 776.3, [$^{PB}$M+H]$^+$) was selected for beam-type CID. Two pairs of C=C diagnostic ions at *m/z* 467/493 and *m/z* 495/521 were observed. These diagnostic ions clearly suggested the existence of Δ9 and Δ11 C=C locations for the C18:1 chain of PE 16:0_18:1, thus identifying the C=C location isomers as PE 16:0_18:1(Δ9) and PE 16:0_18:1(Δ11) (Fig. 2d). Because standards of most lipid C=C location isomers are not commercially available, we could not perform isomer quantitation. Nevertheless, abundance ratios of C=C diagnostic ions of the isomers do have linear correlations with their concentration ratios[22,27], which are used to indicate relative isomer ratios throughout the study. As an example, the relative ratio of PE 16:0_18:1(Δ9) and PE 16:0_18:1(Δ11) isomers were obtained from measuring the abundance ratio of the diagnostic ions (Rel. C$_{Δ9}$/C$_{Δ11}$ = ($I_{467}$ + $I_{493}$)/($I_{495}$ + $I_{521}$)).

It is worth noting that in the workflow of C=C location assignment fatty acyl chain composition needs to be determined first. If there is only one or one dominant of unsaturated fatty acyl composition of the unknown lipid, which is the case for most GPs reported herein, the locations of C=C can be directly ascribed to that fatty acyl. For situations where multiple unsaturated fatty acyl compositions co-exist, the contribution of each isomer is estimated from the relative abundance of product ions from LC–MS/MS in negative ion mode. The more abundant C=C diagnostic ions are ascribed to the major species (Supplementary Figs. 7 and 8). For the minor components if the diagnostic ions have limited S/N or overlap with that of the more abundant species, we choose not to report the C=C location. Examples of C=C assignment of unsaturated PEs, PCs, PGs, and PIs consisting of mono- or polyunsaturated fatty acid (PUFA) chains are shown in Supplementary Figs. 9 and 10. Owing to fatty acyl

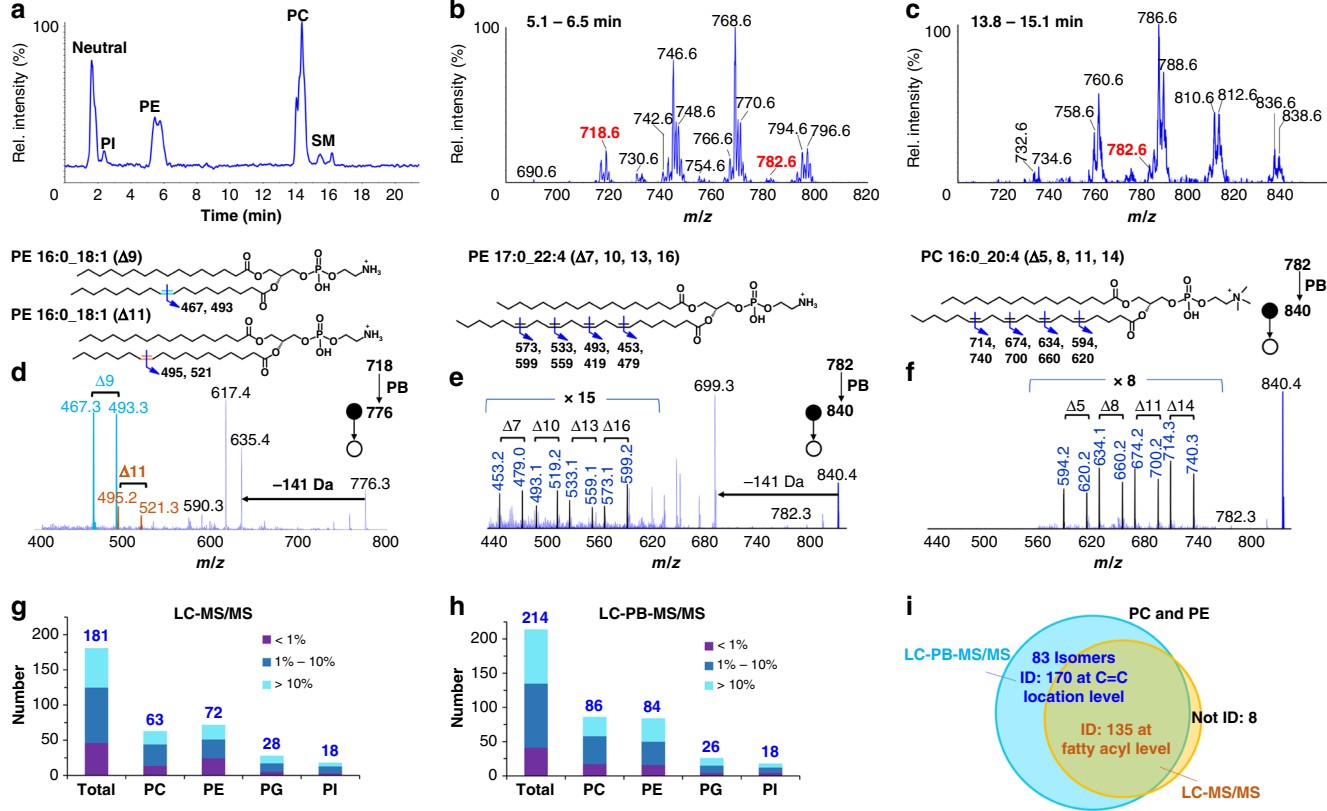

**Fig. 2** Analysis of GPs in bovine liver polar extract. **a** Chromatogram of HILIC separation (0.1 μg lipid per injection). **b** PE profile resulting from 141 Da NLS (RT: 5.1–6.5 min); **c** PC profile resulting from 184 m/z PIS (RT: 13.8–15.1 min). **d** LC–PB–MS/MS of PE 34:1 (RT: 5.7 min, m/z 776.3) reveals the presence of two C=C location isomers, PE 16:0_18:1(Δ9) and PE 16:0_18:1(Δ11). LC–PB–MS/MS of **e** PE 17:0_22:4 (RT: 5.2 min, m/z 840.4) and **f** PC 16:0_20:4 (RT: 14.3 min, m/z 840.4). The number of GP molecular species identified by **g** regular LC–MS/MS and **h** LC–PB–MS/MS. **i** Comparison of unsaturated PC and PE identified by LC–MS/MS and LC–PB–MS/MS

chain analysis before C=C location determination, we also confidently identified a series of GPs containing odd chains, e.g., C15, C17, and C19 with 0–2 degrees of unsaturation (Supplementary Fig. 11). These data led us exclude some other possible isomeric structures, e.g., N-methyl PEs[28]. Ether GPs are isobaric to diacyl GPs, but typically exist at lower abundances[29]. In this study, we only report the ether lipids of which we have gathered confident MS/MS data (Supplementary Data 1). Given the limited mass resolving power from ion trap instrument, we could not confidently identify ether GPs if they co-exist with diacyl GPs as a minor component and therefore they are not reported.

HILIC separation largely reduces ion suppression among different lipid subclasses commonly encountered for complex mixture analysis by MS. We found that the detection sensitivity for PEs from bovine liver extract was increased by 4–7 times, which greatly facilitated structural analysis using PB–MS/MS for PEs of lower abundances (Supplementary Fig. 12). For instance, the Relative Intensity% (Rel. Int.%) of PE 39:4 (monoisotopic molecular mass: 781.5622 Da, m/z 782.6) was less than 2% of the most abundant species in the subclass (PE 38:4, m/z 768.6) (Fig. 2b), while an isomeric PC (PC 36:4, m/z 782.6) also existed in the extract sample (Fig. 2c). These would have had a severe adverse impact on the identification of C=C locations in PE 39:4 if PB–MS/MS analysis was directly applied to the lipid extract. As a contrast, LC–PB–MS/MS offered high quality data for PE 39:4 ([PBM+H]+, m/z 840.4, Fig. 2e), from which four pairs of fragment ions could be picked out as C=C diagnostic ions (m/z 453/479, 493/519, 533/559, and 573/599). Combining information from fatty acyl composition (C17:0 and C22:4), this PE was confidently identified as PE 17:0_22:4 (Δ7, 10, 13, and 16), with

no other major C=C location isomer identified. Similarly, its isomer, PC 36:4, was identified as PC 16:0_20:4 (Δ5, 8, 11, and 14) (Fig. 2f).

Due to the loss in conversion of unsaturated lipids to their PB products, sensitivity of LC–PB–MS/MS is expected to be lower than LC–MS/MS. Nevertheless, from 181 phospholipids identified at fatty acyl level by LC–MS/MS, 86% of them were successfully analyzed for C=C location determination using LC–PB–MS/MS, leading to identification of 214 distinct molecular species (Fig. 2g, h). Because a significant portion of GPs containing C=C location isomers, the molecular species identified by LC–PB–MS/MS are more than those identified at fatty acyl level. Among those, 19% are low-intensity GPs with Rel. Int.% <1% (normalized to the most abundant species in each subclass), 44% with Rel. Int.% in the range of 1–10%, while 37% with Rel. Int.% higher than 10%. These numbers are comparable to GPs identified by LC–MS/MS (25%, 44%, 31% for Rel. Int.% <1%, 1–10%, and >10%, respectively), suggesting that the LC–PB–MS/MS system maintains adequate sensitivity for lipidomic analysis. Using PC and PE as an example, 94% of them were determined with C=C location specificity by LC–PB–MS/MS (Fig. 2h, i). Among those, 40 lipids identified at fatty acyl level were found to have two or more C=C location isomers, leading to a total of 170 PCs and PEs identified at C=C location level.

Our group and others have shown that fatty acyl C18:1 typically contains C=C location isomers at Δ9 (major) and Δ11 (minor) for lipids analyzed from mammalian cells, animal tissue, and human plasma[17,19,22]. This C=C location isomer composition was consistently observed for all GPs containing C18:1 from

bovine liver extract (complete list shown in Supplementary Data 1). Interestingly, for PE 18:1_18:1 and PC 18:1_18:1, we also detected relatively low intensity C=C diagnostic ions that could be assigned as the Δ10 isomer, although other unusual structural isomer may also be possible (i.e. branched fatty acyl (Me-C17:1 (Δ9)). For fatty acyl C16:1, Δ7 (minor, has been reported previously by GC–MS[30]) and Δ9 (major) isomer pair was detected. Moreover, C=C location isomers of polyunsaturated fatty acyl chains from a series of PC and PE were also identified using LC–PB–MS/MS. For simplicity, the omega (ω) nomenclature is used for PUFA annotation; the lipid isomers include ω-6 (major) and ω-9 (minor) isomer pairs in C18:2 and C20:2, and ω-3 and ω-6 isomer pairs in C18:3, C20:3, C22:5 (Supplementary Fig. 9, Supplementary Data 1). Notably, 48 lipid species that are not listed in LIPID MAPS database for C=C locations, have been identified (highlighted in Supplementary Data 1). LC–PB–MS/MS was successful in analyzing unsaturated free FAs, PG, and PI in negative ion mode. Although the PB products of unsaturated glycerolipids were detected ([PBM+NH₄]⁺ or [PBM+Na]⁺), the online PB reaction needs to be tailored for these neutral lipids.

**Analysis of human breast cancer tissue samples.** In a previous study, a shotgun lipid analysis incorporated with the PB reaction revealed significant changes in the C=C isomer compositions of C18:1 (Δ9/Δ11) for PC 18:0_18:1 and PC 18:1_18:1[22]. These results suggest the potential of monitoring the change of lipid C=C location isomers in discovery of biomarkers, which would otherwise be invisible if using conventional lipid profiling techniques. To evaluate the developed LC–PB–MS/MS method for large-scale screening of lipid C=C location isomers for biomarkers discovery, polar lipid extract from human breast tissue samples was analyzed, including 6 cancerous and their para-carcinoma sections for comparison (equivalent to lipid extraction from 0.14 mg tissue per LC injection, 3 technical repeats). A total of 143 unsaturated PEs (71) and PCs (72) molecular species were

identified at C=C location level. Besides the most commonly observed Δ9/Δ11 isomer pair in C18:1 and ω-3/ω-6 isomer pair in C18:3, other less common isomer pairs were also detected, including Δ7/Δ9 isomer pair in C16:1, and ω-3/ω-6 isomer pair in C18:2 (Supplementary Fig. 13). A complete list of the identified GPs is documented in Supplementary Data 2.

Relative quantitation of PCs and PEs at subclass level was performed for the normal and cancerous tissue samples (Fig. 3a, b, and Supplementary Figs. 14 and 15). Only six lipid species (PC 32:1, PC 34:2, PC 34:1, PE 36:1, PE 38:4, and PE 40:7) out of 109 identified PCs and PEs were found to have statistically significant changes ($P < 0.001$, determined by two-tailed student's $t$ test, average RSD: $27 \pm 12\%$, mean ± s.d., $N = 6$). This result is consistent with literature reports, where PE 34:1, PE 36:2, PE 38:4, PC 32:1, and PC 36:2 have been identified to have significant changes in breast cancer cell lines or tissue samples[31–35].

Given the ubiquitous existence of Δ9/Δ11 isomer pair in all C18:1-containing lipids, compositional analysis was performed for 29 pairs of PC and PE isomers from normal and cancerous breast tissue samples (Supplementary Fig. 16). The $t$ test analysis found twelve isomer pairs exhibiting significant changes ($P < 0.001$, average RSD = $16 \pm 4\%$) in Δ9/Δ11 composition, including six PC pairs and six PE pairs (Fig. 3c, d and Supplementary Fig. 17). To correct false discovery, multiple testing was performed[36]. Significant changes were still observed for these lipid isomers, except for PC 19:0_18:1 (corrected value of −0.0005). Among those, only two overlapped with the ones exhibiting significant changes at subclass level analysis, i.e. PC 16:0_18:1 and PE 18:0_18:1. The improved differentiating power from isomer analysis largely benefited from the capability of quantifying changes from each individual C=C location isomer. For instance, C=C isomer compositional analysis revealed that PE 34:1 consisted of two C=C location isomers, PE 16:0_18:1(Δ9) and PE 16:0_18:1(Δ11) (Fig. 3d). While the total abundance of the two isomers did not show significant changes as measured at subclass level ($I/I_{IS} = 0.17 \pm 0.03$ in normal vs. $0.21 \pm 0.04$ in

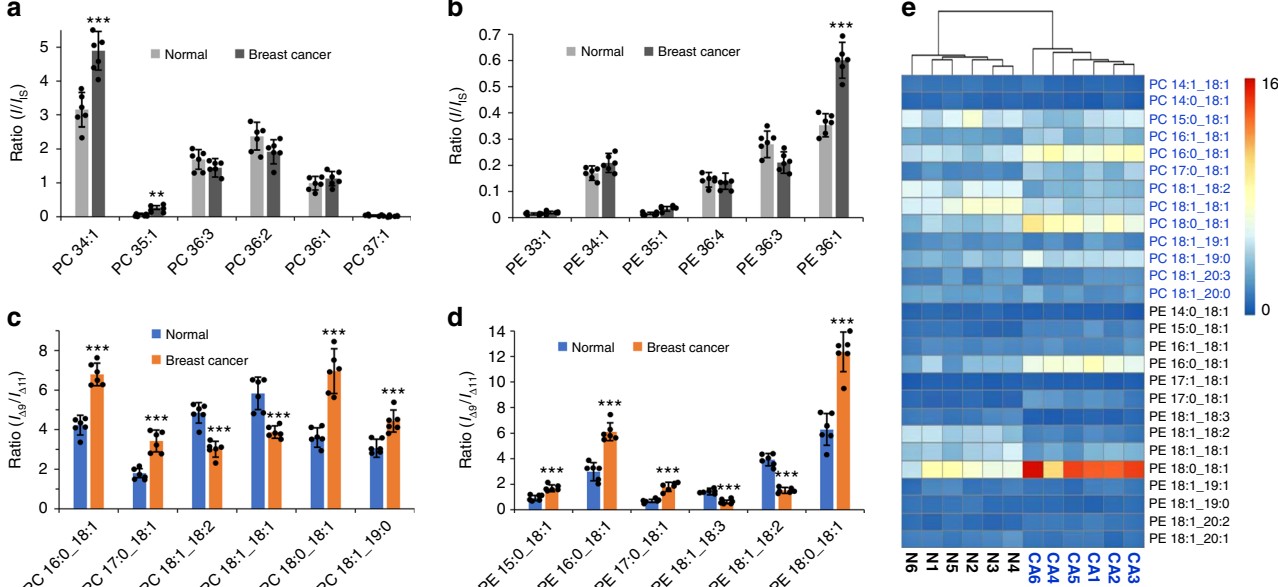

**Fig. 3** Analysis of GPs containing C18:1 Δ9/Δ11 isomers in human breast cancer samples. Relative quantitation of **a** PCs and **b** PEs at subclass level between normal and cancerous breast tissue samples by LC–MS/MS (PC 15:0/15:0 and PE 15:0/15:0 used as internal standards). Relative isomer compositional analysis of C18:1 Δ9/Δ11 for **c** PC and **d** PE between normal and cancerous breast tissue samples by LC–PB–MS/MS. Differences between the two groups of samples were evaluated for statistical significance using the two-tailed student's $t$ test (*$P < 0.05$, **$P < 0.01$, ***$P < 0.001$). Each value represents the mean ± s.d. ($N = 6$). **e** Hierarchical cluster analysis based on C18:1 Δ9/Δ11 ratios of 29 pairs of PC and PE isomers in normal (N1–N6) and cancerous breast tissue samples (CA1–CA6). Colors represent different ratios as indicated by the color bar

cancerous tissue samples, Fig. 3b), their ratios have changed significantly from normal ($I_{\Delta 9}/I_{\Delta 11} = 2.99 \pm 0.7$) to cancerous tissue samples ($6.11 \pm 0.7$).

Hierarchical cluster analysis was further employed to analyze PC and PE compositions. Data analysis based on relative quantitation ($I/I_{IS}$) at subclass level failed in correctly grouping the normal and cancerous samples (Supplementary Fig. 18). As a contrast, cluster analysis of the 29 pairs of C=C location isomers correctly grouped the 12 tissue samples into normal vs. cancerous (Fig. 3e). Obviously, relative quantitation of C18:1 C=C location isomers for a wide variety of lipid species adds important capability for study of pathological states of tissues. It is worth noting that the relative ratios of $\Delta 9/\Delta 11$ isomers of free FA 18:1 did not show significant changes between the normal ($8.9 \pm 0.7$, $N = 6$) and cancerous tissue samples ($7.4 \pm 0.5$, $N = 6$) (Supplementary Figs. 19 and 20), which were also close to that obtained for human plasma samples (in the range of 9–11). The above results suggest that although the free FAs and the corresponding fatty acyls in phospholipids may contain the same pair of C=C location isomers, their ratios can be quite different or even unrelated.

**Analysis of plasma samples of T2D patients.** Homeostasis of plasma lipidome is closely related to the etiology of T2D[37,38]. Several plasma lipids have been suggested as potential biomarkers for T2D risk assessment[39,40]; and C=C location isomers of certain TAGs have been found to change significantly in T2D plasma[41]. For plasma lipid biomarker discovery, large sampling sizes, ranging from tens to over 10,000 human samples, are needed to overcome high degrees of patient-to-patient variations commonly found in clinical blood plasma samples[37–40]. Our earlier findings from breast cancer tissue lipid analysis showed that the composition changes of lipid C=C location isomers could be more reliable in terms of reflecting pathological changes occurred to a biological system than lipid profiling at class level. It is of interest to investigate if it is possible to employ relatively small sampling size for the discovery of potential markers based on monitoring lipid C=C location isomers. In order to explore

this potential, we applied LC–PB–MS/MS method for the analysis of human plasma samples collected for T2D studies, with only six samples for diseased and control, respectively.

Relative quantitation of PEs and PCs was first conducted at subclass level for T2D plasma ($N = 6$) and normal control ($N = 6$) samples. No significant difference in lipid composition was identified at subclass level, which could be expected with a sampling size as small as six (Supplementary Figs. 21 and 22). It is worth noting that by using a much larger sampling size, i.e., 70 plasma samples, previously Wang et al.[39] discovered significant changes at subclass level for PE 38:4 and PE 38:6 in T2D plasma samples.

Analysis at C=C location level identified a total of 116 distinct PC (74) and PE (42) molecular species (complete identification list in Supplementary Data 3). Consistent with previous findings, all GPs containing C18:1 chain were found to be a mixture of $\Delta 9$ and $\Delta 11$ C=C location isomers (Supplementary Data 3). For the rest of GPs containing unsaturated fatty acyls, either only one dominant C=C location isomer was detected or the C=C location isomers were only discovered from a limited number of lipids. Some C=C location isomers, previously not discovered in tissue samples, were also identified, such as $\Delta 10$ vs. $\Delta 12$ isomers for PC O-16:1/16:0 and ω-6 vs. ω-9 isomer pair of C18:2 in PC 15:0_18:2 (Supplementary Fig. 23).

Relative isomer ratios were measured for 19 pairs of $\Delta 9/\Delta 11$ C=C isomers from PCs/PEs containing C18:1. Based on these values, the twelve plasma samples could be classified correctly into normal and T2D from hierarchical cluster analysis (Fig. 4a), while lipid profiling analysis failed to do so (Supplementary Fig. 24). More importantly, seven distinct isomer pairs were identified to exhibit significant changes in relative isomeric ratios between T2D and normal plasma samples, including two PEs and five PCs ($I_{\Delta 9}/I_{\Delta 11}$, $P < 0.001$, determined by two-tailed student's $t$ test, Fig. 4b and Supplementary Fig. 25). There is no obvious trend in the changes of the $\Delta 9/\Delta 11$ isomers and neither do the changes correlate with relative isomeric composition in free FA 18:1 ($I_{\Delta 9}/I_{\Delta 11} = 10.5 \pm 0.9$ in control, $N = 6$, and $11.4 \pm 0.9$ in T2D, $N = 6$) (Supplementary Figs. 26 and 27). For instance, the relative isomeric ratio ($I_{\Delta 9}/I_{\Delta 11}$) in PE 16:0_18:1 decreased from $5.0 \pm 0.5$ in control to $3.7 \pm 0.3$ in T2D, while the ratio of PC

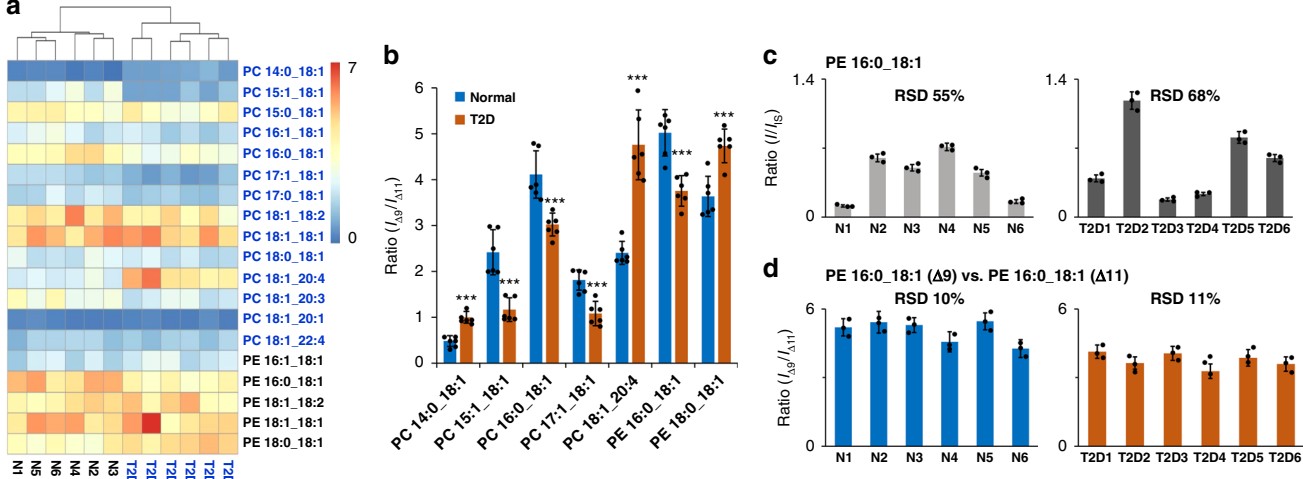

**Fig. 4** Analysis of GPs containing C18:1 $\Delta 9/\Delta 11$ isomers in human plasma samples. **a** Hierarchical cluster analysis of $I_{\Delta 9}/I_{\Delta 11}$ in normal (N1–N6) and T2D plasma samples (T2D1–T2D6) provides correct grouping of the two sets of samples. Colors represent different ratios as indicated by the color bar. **b** Seven pairs of lipid isomers (C18:1 $\Delta 9/\Delta 11$) are found to exhibit significant changes in relative composition ratios between normal control and T2D. Differences between the two groups were evaluated using the two-tailed student's $t$ test (***$P < 0.001$). Each value represents the mean ± s.d. ($N = 6$). **c, d** Comparisons of individual-to-individual variations based on **c** relative quantitation ($I/I_{IS}$) of PE 16:0_18:1 and **d** relative isomer ratio ($I_{\Delta 9}/I_{\Delta 11}$) measurement of PE 16:0_18:1 in normal and T2D plasma samples. Each value represents the mean ± s.d. ($N = 3$). Precision of relative isomer ratio measurements is much less affected by individual-to-individual variations as compared to relative quantitation measurements at subclass level

18:1_20:4 increased from 2.4 ± 0.4 in control to 4.8 ± 0.9 in T2D. These data suggest that the metabolic processes associated with these isomeric species responded quite differently under the T2D conditions.

Biomarker discovery using blood/plasma samples is attractive for the development of disease diagnosis or preventative therapies due to its relatively low invasiveness for sampling. As a preliminary exploration demonstrated here, lipid C=C location isomer composition is clearly much more reliable to reflect the T2D conditions than the change of lipid composition at subclass level, due to significantly reduced individual-to-individual variations. Using relative quantitation of PE 16:0_18:1 as an example, RSDs from three technical repeats of one individual are all round 8% for normal control and T2D samples (Fig. 4c), which is within the typical variation of LC–MS/MS experiments. However, individual-to-individual variation is huge at lipid subclass level, leading to RSD of 55% (normal control, $I/I_{IS} = 0.4 ± 0.2$, $N = 6$) and 68% (T2D, $I/I_{IS} = 0.6 ± 0.4$, N = 6) for relative quantitation of PE 16:0_18:1. When counting Δ9/Δ11 relative isomer ratios for PE 16:0_18:1, the RSDs (8%) of the technique repeats are quite comparable to that of subclass quantitation; however, individual-to-individual variation is significantly reduced, at 10% for normal control ($I_{Δ9}/I_{Δ11} = 5.0 ± 0.5$) and 11% for T2D ($I_{Δ9}/I_{Δ11} = 3.8 ± 0.4$). This aspect leads to a confident correlation of relative changes in Δ9/Δ11 isomer compositions to T2D condition.

In fact, small RSDs, typically within 15%, were found for all Δ9/Δ11 isomeric ratios measured for C18:1, i.e., (14 ± 4%) for normal and (15 ± 4%) for T2D. As a contrast, RSDs of 43 ± 14% for normal and 42 ± 12% for T2D were obtained for lipid profiling at subclass level. Large variations of lipid concentrations in blood is expected because they are heavily influenced by individual differences in circadian rhythm, temporal metabolism variance, and several other factors[42,43]. The finding of high precision in measuring relative ratio of lipid C=C location isomers suggests tighter regulation/dysregulation at the isomer level which is unaffected by sample complexity or preparation procedures. These results are also consistent with recent isomer-resolved imaging MS studies which have highlighted tissue-tissue and tissue-tumor differences at the isomer level that are invisible at the molecular lipid level[44,45].

## Discussion

In this work, we have successfully incorporated the PB–MS/MS technique into LC–MS workflow that has been widely employed for large-scale lipid analysis. This step of development is critical for lipidomics studies since now the structural information related to C=C locations in lipids becomes readily available for understanding the lipidome and associated biological systems. Using polar lipid extract from bovine liver as a benchmark test, more than 200 unsaturated phospholipids have been identified for the sites of unsaturation, among which 55 groups of C=C location isomers have been discovered, including 48 molecular structures which were not predicted by LIPID MAPS database. Undoubtedly, these data set a record for comprehensive structural analysis of phospholipids and demonstrate that the LC–PB–MS/MS system readily meets the needs in lipidomic analysis in terms of throughput, sensitivity, and lipid coverage, while providing structural analysis at a deeper level. Streamlined data acquisition and data dependent analysis procedure are also key to the adoption of the LC–PB–MS/MS platform for large-scale lipid analysis. The home-developed software (Lipid Omega Analyzer) is our first step toward this direction, which is incorporated with the functions for guiding PB–MS/MS and performing multilevel structural analysis for phospholipids including subclass, acyl/alkyl chain, de novo C=C location identification.

With the workflow we developed in this study, identification and quantitation lipid C=C location isomers, lipid biomarker discovery now will no longer be limited to profiling at subclass level, which is blind to individual changes of isomers. The correlation between the composition changes of the C=C location isomers and the status of the biological system could serve well for biomarker discovery. Analysis of human breast cancer tissue samples reveals that relative composition changes of C18:1 Δ9 and Δ11 isomer pairs in PCs and PEs provide reliable parameters for tracing pathological conditions. Twelve out of twenty-nine isomer pairs have been identified to exhibit significant changes in isomer composition; among those only two are overlapping with the six species (out of one hundred and nine PC and PE molecular species) identified from subclass level analysis.

Analysis of human T2D plasma samples demonstrates another exciting aspect of screening C=C location isomer composition changes for biomarker discovery. Compositions of seven isomer pairs (C18:1Δ9/Δ11 in PCs and PEs) have been observed with significant changes, while subclass level profiling failed in identifying any significant difference due to small sampling size used. The high sensitivity in detecting changes in lipid C=C location isomers is a direct result of high measurement precision, which is not affected by variations among individuals. Our analysis results show that RSDs from measuring relative isomer ratios are significantly smaller than relative quantitation from subclass level (15% vs. 42%) for analyzing plasma lipids. This is consistently observed for breast cancer tissue analysis, although the difference in RSDs (16% in isomer analysis vs. 27% in subclass analysis) is less dramatic as found in human plasma. It is worth noting that these high precisions of C=C location isomer ratio measurements were obtained without requiring use of IS. This aspect represents a huge advantage for clinical and point-of-care analysis, where small sample volumes are used and simple operation is highly appreciated.

## Methods

**Sample preparation**. Lipid standards and bovine liver polar extract were purchased from Avanti Polar Lipids, Inc. Lipid stock solutions were prepared in chloroform or methanol and diluted to working concentrations by methanol.

Human breast cancer tissue samples and plasma samples were supplied by the specimen bank of Dongfeng Hospital of Hubei University of Medicine (sample information was listed in the Supplementary Data 4). All the procedures related to these samples were compliant with all relevant ethical regulations set by the Ethical Review Board of Tsinghua University (IRB No. 2017007). Informed consent was obtained from all participants. A modified Folch method was employed for lipid extraction from tissue and plasma samples. For lipid extraction from tissue samples, 70 mg of tissue sample was placed in a 10 mL-centrifuge tube with 1 mL of deionized water added. The tissue sample was homogenized by a handheld homogenizer (Jingxin Technology) at 40,000 Hz for 5 min and subsequently mixed with 1-mL methanol and 2 mL chloroform for liquid–liquid extraction. After 5 min-vortex, the mixture was centrifuged at 11,269×g for 8 min (Eppendorf). The bottom layer was collected and transferred to a 10 mL-glass test tube. The above extraction process was repeated once. The chloroform layers from the two extractions were combined and dried under nitrogen flow. The extract was reconstituted into 1-mL methanol and stored at −20 °C before analysis. For lipid extraction from plasma samples, 50 μL plasma was diluted by 1 mL deionized water in a 10 mL-centrifuge tube and followed by an addition of 1 mL methanol and 2 mL chloroform. After 5 min-vortex, the mixture was centrifuged at 11,269×g for 8 min. The extraction was repeated once. The chloroform layers from the two extractions were combined and dried under nitrogen flow. The extract was reconstituted into 1-mL methanol and stored in −20 °C before analysis.

**LC–PB–MS platform**. The LC–PB–MS system consists of an ExionLC AC system (Sciex), a 4500 QTRAP mass spectrometer (Sciex) and a home-built flow micro-reactor. The LC is equipped with a degasser, two pumps, an automatic sampler, and a column oven. Separation was performed on a HILIC column (150 mm × 2.1 mm, silica spheres, 2.7 μm) (Sigma-Aldrich). The column temperature was set at 30 °C. Mobile phase consisted of A: acetone/ACN (50/50, $v/v$) and B: ammonium acetate aqueous solution (10 mM). Gradient elution was applied for separation (A started from 90%, decreased to 85% at 5 min, then decreased to 80% at 8 min, kept at 80% within 8–15 min, and decreased to 70% in 16 min and kept this percentage to 20 min) at a flow rate of 0.2 mL min$^{-1}$.

MS analysis was performed on the 4500 QTRAP triple quadrupole/linear ion trap hybrid mass spectrometer, employing data collection modes of PIS, NLS, enhanced MS, and enhanced product ion (EPI). Unless otherwise specified, parameters of MS were set as following: ESI voltage (±4500 V), curtain gas (35 psi), interface heater temperature (400 °C), nebulizing gas (GS1, 30 psi), and nebulizing gas (GS2, 30 psi). Other parameters for each data collection mode were listed in the Supplementary Data 5.

FEP tubing (0.03-in. i.d., 1/16-in. o.d., Zeus Inc.) was concentrically coiled at a diameter of 2.5 cm on a rack which was made of four stainless-steel pillars and two pieces of plastic pedestals. The inlet and outlet of the FEP coil was fixed onto the pillars by stainless-steel nuts and connected with LC flow path via standard polyether ether ketone (PEEK) finger-tight fittings (1/16-in. i.d.). A low-pressure mercury lamp with emission centered around 254 nm (Model No. 80-1057-01, BHK Inc.) was fixed at the center of the FEP coil though a hole in the top pedestal. The whole setup was covered with aluminum foil to prevent straylight outside the reactor. The inlet and outlet of the microreactor was connected to the LC column and the ESI source via PEEK tubing (0.01-in. i.d., 1/16-in. o.d.).

**Data collection via LC-MS/MS.** The UV lamp was kept off for LC–MS/MS analysis. NLS and PIS were performed for lipid subclass profiling. For PE and PC identification, the MS analysis was performed in positive ion mode, the method set as NLS of 141 Da (PEs) in the retention time of 10 min, and PIS at $m/z$ 184 (PCs) in the RT of 10–20 min. The collision energy was set at 40 eV for NLS and 42 eV for PIS. PIS at $m/z$ 153 was performed for profiling PGs and PIS at 241 was performed for profiling PIs in the negative ion mode. The collision energy was set at 45 eV. A list of precursor ions for chain analysis and subsequently a list of PB precursor ions of unsaturated lipids was generated from NLS and PIS for each lipid extract from tissue or plasma samples. For chain analysis, EPI mode was used for MS/MS analysis in negative ion mode. PGs and PIs were analyzed in 0–5 min, PEs were analyzed in the first 10 min and PCs were analyzed in 10–20 min. The collision energy was set at 40–45 eV.

**Data collection via LC–PB-MS/MS.** The low-pressure mercury lamp (254 nm) was prewarmed for at least for 5 min before use. For PCs and PEs, a list of the PB precursor ions of unsaturated lipids was generated based upon the results from the previous LC–MS/MS (NLS 141 and PIS 184) process. The list was input into the console of the mass spectrometer, along with parameters including $m/z$ range ($m/z$ 350–1000 for PEs, $m/z$ 500–1000 for PCs) and collision energies (40 eV for all PE precursor ions, and 42 eV for all PC precursor ions). The parallel EIP mode in positive ion mode was applied for LC–PB-MS/MS analysis. The EPI list was separated into several sections according to the retention times of different lipid species. For PG and PI, MS[3] analysis in negative ion mode was applied. PB precursor ions of unsaturated lipids were produced based upon the results from the previous LC–MS/MS process (PIS 153 for PGs and PIS 241 for PIs, first precursor, +58 Da) and LC–MS/MS (EPI) in negative ion mode (second precursor, +58 Da), along with parameters including $m/z$ range ($m/z$ 100–400 for PIs and PGs) and collision energies (45 eV for beam-type CID and 0.1 (arbitrary units) for ion trap CID). For FA 18:1, the EPI mode in negative ion mode was applied for LC–PB-MS/MS analysis. The precursor ion was set as $m/z$ 339.3, the collision energy was set as 45 eV for beam-type CID and 0.1 (arbitrary units) for ion trap CID.

**Data analysis.** Data analysis was partially done by the Analyst software equipped with the QTRAP 4500, by which mass spectra can be generated and peak lists can be exported. An in-house developed data analysis software, LOA, was employed for comprehensive lipid structure identification. Proteowizard was used to convert .wiff data from the Analyst to an open file format (.mgf), which can be read by the LOA. The database from LPIDMAPS was used for generating possible lipid structures at fatty acyl compositional level. From reading the PIS and NLS data from LC–MS/MS, LOA extracted a list of lipids, including retention time, precursor $m/z$, molecular mass, and subclass and such a list was input into the Analyst software to guide LC–MS/MS in negative ion mode for fatty acyl chain analysis. LOA read the data from LC–MS/MS in negative ion mode and provides lipid identification at fatty acyl (alkyl) chain level.

LOA conducted de novo analysis of the data generated from LC–PB–MS/MS and the C=C location of each unsaturated acyl (alkyl) chain was assigned based on the detection of C=C diagnostic ion pairs. LOA also calculated the intensity ratios of diagnostic ions corresponding to C=C location isomers from for relative isomer quantitation. Details about LOA are shown in the Supplementary Figs. 4–7. With proper information added in the database for internal standards and corresponding calibration curves, the capability of LOA can be extended for absolute quantitation.

**Reporting summary.** Further information on experimental design is available in the Nature Research Reporting Summary linked to this article.

**Code availability.** The LOA software and three representative test datasets are available on Github (https://github.com/LipidAnalysis?tab=repositories). The code of the LOA is available from the corresponding authors upon reasonable request.

## Data availability

A reporting summary for this Article is available as a Supplementary Information file. All data supporting the findings of this study are available from the corresponding authors on reasonable request.

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

## Acknowledgements

This work was supported by National Natural Science Foundation of China (Project Nos. 21722506, 21621003 and 21627807) and National Institutes of Health of USA (Project R01GM118484). The authors thank Jiexun Bu, Gang Bai, Xueye Zhang, and Chunchao Shang for support in developing the data analysis software.

## Author contributions

W.Z., Z.O., and Y.X. designed research and wrote the paper. W.Z., D.Z., Q.C., and J.W. performed research and analyzed the data. D.Z., W.Z., and Y.X. developed data analysis software and algorithms. All authors discussed the results and edited the manuscript.

## Additional information

**Competing interests:** Z.O. is the founder of PURSPEC Technologies Inc. The remaining authors declare no competing interests.

