## [Peer Review File · Nature Communications]

Reviewers' comments:

Reviewer #1 (Remarks to the Author):

Xia and co-workers have pioneered the application of the Paterno-Buchi reaction for the assignment of carbon-carbon double bond position in simple and complex lipids and have previously used the technique to demonstrate variation in lipid isomer abundances associated with a number of pathologies; notably cancer. In this manuscript this novel analytical workflow has been fully integrated into an impressive online, high-throughput LC-MS/MS methodology; achieving broad lipidome coverage. Furthermore, a data-processing software has been introduced to rapidly identify lipid isomers in complex mixtures and extract relative quantitation of ions representative of isomeric forms. This integration, and the large data-sets it has facilitated, are significant for two reasons. Firstly, the lipidomes of key systems (e.g., plasma, bovine liver and cancer tissue) are shown to be significantly expanded through the ability of this protocol to identify isomeric contributors; effectively splitting single molecular lipid assignments (that can be achieved by conventional mass spectrometric analysis) into multiple isomers differing in their site(s) of unsaturation. Secondly, the comparison of sample cohorts for cancer and type-2 diabetes reveals (for the first time) that the regulation/dysregulation of lipid metabolism can be much more sensitively probed at the isomer level. This is effective because tissue heterogeneity (particularly in cancer tissue) can make normalization of lipid abundance very difficult. What should one normalize too? Standard measures such as tissue mass, total DNA and total protein are all be significantly affected by sample heterogeneity and thus makes comparison of absolute lipid abundances between samples quite challenging. In contrast lipid isomers share common detection efficiency and thus changes in their relative abundance are shown to be a sensitive probe for metabolic change. This demonstration is important as it paves the way for the use of lipid isomer ratios as robust diagnostics against the well documented natural variations in tissue and plasma resulting from diet, exercise, medication, diurnal rhythms etc. These findings are impactful for a wide audience in lipid biochemistry and translational biomedical sciences and I strongly support their publication once the authors consider the following corrections/suggestions below.

1. The relative quantitative of isomers, described on page 9, is based on product ion abundance ratios. To my knowledge this has not been independently validated for the intact glycerophospholipids, i.e., dissociation of $[PC + 58 + H]^+$, $[PE + 58 + H]^+$ etc. As presented the implication is that there is no PB reaction bias or dissociation effects associated with double bond position. This assumption is inconsistent with prior reports from this group (e.g., Ma et al. PNAS 2016, ref [22]) that show calibration curves derived from fatty acid standards $[M-H]^-$ anions deviate significantly from a slope of 1. Further evidence needs to be provided therefore that these relative product ion abundances reflect relative isomer concentrations. If they do not, which I suspect is the case, this is not critical as the important result is visualisation in the change in relative abundance NOT the relative abundance itself. Nonetheless the terminology used needs to clearly reflect the evidence.

2. Pg 10 and elsewhere the term MW or molecular weight is used. MW has units of g/mol and is not relevant to the mono-isotopic measurements undertaken in a mass spectrometer. Molecular mass or mass-to-charge ratio should be applied depending if the discussion pertains to neutrals or ions.
3. The analysis of both plasma, tissue and bovine liver lipidomes reports a number of odd chain species. E.g., pg. 11 PE 17:0_22:4. The authors should articulate how these are differentiated from isobaric ether-linked species (e.g., in this case PE(O-40:4)) or mono-methyl and dimethyl PE species, e.g., Me-PE38:4 (known to be present in liver – see Ejsing and co-workers *Biochimica et Biophysica Acta* 1811 (2011) 1081–1089). There may well be retention time information or negative ion CID product ions that supports the assignments made but this needs to be demonstrated/tabulated for cases where unusual or hitherto un-reported lipids are presented. This is important, as some of these samples have previously been heavily scrutinised and , for example, there is no consensus on odd chain contributions to the phospholipidome in plasma see (Quehenberger *Journal of Lipid Research* Volume 51, 2010 3299). More probably, in this instance at least the putative odd-chains are ether lipids.
4. Similar to (3) where unusual double bond locations are reported it should be clear how these have been assigned to particular chains. E.g., pg 13. A delta-7/delta-9 pair are assigned to 16:1 in a PC 14:0_16:1 but how is the possibility of a contribution from PC 14:1_16:0 excluded?
5. Another example, is the putative delta-10 18:1 (18:1, n-8) on pg 12. I haven't found any mention of this fatty acid in the literature – what would the desaturase be? Are there literature (or other) GC data to support unassigned 18:1 species? It could be a branched chain Me-17:1(n-7) or alternatively the signal might arise from a contribution of a methylPE MPE(35:2). MPE and DMPE are known to be present in liver. Bilgin et al. *Biochimica et Biophysica Acta* 1811 (2011) 1081–1089
6. Pg 15. The double bond isomer ratios of phospholipids are compared to free fatty acids. Some more detail on the method for FFA determination should be provided. These methods are reasonably controversial and are subject to un-intended hydrolysis and background fatty acid contamination. For example, the solvents and glassware likely carry a significant abundance of oleic acid (it's always in my mass spectra!) this would likely skew the delta-7/delta-9 ratio reported unless there is careful examination of the background contribution in blank injections.
7. Isomer-level discrimination for Type-2 diabetes has been previous demonstrated by Stahlman et al. 2012 *Diabetologia* (DOI 10.1007/s00125-011-2444-6). This was only shown for a small set of isomers but it is nonetheless satisfying to see these differences in regulation of unsaturation mirrored across the lipidome.
8. I could not locate Supplementary Figure 24 that is cited on pg 17.
9. I could not located key supplementary Tables.
10. Pg 17. Figure caption states "Precision of isomer ratio measurements is much less affected by subject variations as compared to relative quantitation measurements." This is a very clear articulation of a critical point. To some degree it is lost in the figure caption and deserves a bit more discussion. This finding suggests tighter regulation/dysregulation at the isomer level which is unaffected by dilution or other normalization complexity (discussed above). It is also consistent with

recent isomer-resolved imaging MS studies which have highlighted tissue-tissue and tissue-tumour differences at the isomer level that are invisible at the molecular lipid level.

<https://doi.org/10.1002/anie.201806635> and <https://doi.org/10.1002/anie.201802937>

11. Finally, the demonstrated efficacy of double bond positional isomers as biomarkers shown here sets a clear challenge for the mass spectral discrimination of other types of lipid regio- and stereo-isomers.

12. I found some of the English language in the introduction a bit hard to read. I have suggested a few edits as annotations in the attached PDF.

Reviewer #2 (Remarks to the Author):

This is a generally well-written manuscript in which the authors, continuing their previous work, demonstrate that their analytical platform can provide comprehensive and detailed information about the location of double bonds in the fatty acyl chains of PC and PE phospholipids. The analytical approach expands the classic workflow for the determination of phospholipid species (including a liquid chromatography step to separate phospholipid classes, a positive ion analysis to confirm the polar head of phospholipid and a subsequent negative analysis to characterize the fatty acid composition of the molecules) with the inclusion of an online derivatization step of the fatty acid double bonds based on the Paternò-Büchi reaction. This derivatization provides unique information about the position of the double bonds in the fatty acyl chains of phospholipids when they are fragmented in positive ion mode. The inclusion of the derivatizing agent acetone in the mobile phase and the post-column derivatization is both clever and convenient. It is also worth noting that an in-house software, Lipid Omega Analyzer, was developed to help with the analysis of data. As proof-of-concept studies, Zhang et al., use the platform to characterize the phospholipidome of human breast cancer tissue and plasma from patients suffering type 2 diabetes with significant success.

Even though the Paternò-Büchi reaction for the characterization of fatty acid positional isomers has been introduced before by this group and others, although its online incorporation as part of the analytical methodology represents a significant accomplishment, yielding results never achieved before for the comprehensive analysis of individual phospholipid species.

Although the paper provides a well-designed approach for the full characterization of some phospholipid species, some examples of the analysis of other major phospholipid classes in mammals (PI and PS, especially), as well as other minor phospholipid species that play a role in human physiology (PA, PG, BMP and CL) would be useful.

All in all, there are no major concerns that preclude the publication of this article, however, some minor issues should be correctly address before.

Minor concerns:

1. The title does not take advantage of the content of the manuscript. Especially the fact of describing this approach as suitable for “biomarker discovery”. It is true that the ratios between different positional isomers can provide a new layer of information for the discovery of biomarkers. However, pointing it out in the title may be misunderstood by readers as a specific approach for biomarker discovery and thus, reduce the potential of this approach for other biomedical applications that are nowadays at the forefront of lipid science. For instance, positional isomers of palmitoleic acid, sequentially measured using time-consuming GC/MS approaches have shown different effect modulating the inflammatory response (doi: 10.1194/jlr.M079145). With this approach, all 3 isomers could have been identified, quantified and tested for biological effect as part of the same study, considerably reducing time and costs. Other example is the measurement of omega-3 and omega-6 docosapentaenoic acid isomers (known to have different biological effects) esterified in phospholipids at the same time.

The authors are encouraged to consider changing the title of the manuscript to other either more general (“Comprehensive Analysis of Unsaturated Phospholipid Isomers for New Biomedical Applications” or similar) or more focused in the analytical approach (“Comprehensive Analysis of Fatty Acids Positional Isomers of Phospholipids Using an Online Photochemical Derivatization” or similar). A change in the title could make the paper more attractive for readers and increase its scope. As a side comment, the analysis is constrained to PC and PE species, the inclusion of the term “Phospholipids” does not fit with the content of the manuscript.

2. Even though the analysis of other phospholipid classes (PI, PS, PA, PG, BMP and CL) is sometimes mentioned in the text, MS/MS spectra for these lipid classes are not shown. Authors are requested to show at least one MS/MS spectrum of the identification of positional isomers of fatty acids of some of these phospholipid classes. Since the analysis is carried out in positive mode, at least one example of PS and PG molecules should be found. While the ionization in positive mode for PI, PA and CL is poor, the authors are encouraged to find one example of these molecules (for instance the major PI species PI(18:0/20:4)). The inclusion of more examples could generalize the model for the analysis of the entire phospholipidome.

3. Related with the previous concern, ether-linked fatty acids of PC and PE species are not shown in the analysis. This is especially relevant for PC and PE species, where alkyl or vinyl-ether linked fatty acids represent an important part of these phospholipid pools (in PE can represent up to 50% of total mass) and whose dysregulation is related to several diseases (<https://doi.org/10.1016/j.bbadis.2012.05.014>). Is there any difference in the fragmentation behavior of ether-linked fatty acids compared to their ester-linked counterparts for the identification of double bonds position. Please, show a comparative MS/MS spectra of PC or PE species containing the same unsaturated fatty acid in the sn-1 position, but linked with an ester or an ether bond.

4. A recent paper describing the Paternò-Büchi reaction to determine the position of double bonds in free fatty acids has been published (10.1021/acs.analchem.7b02375). Since this paper uses negative ion mode to characterize the fatty acids and discuss about mechanistic aspects of the fragmentation, the authors should cite it, as well as, to discuss briefly the use of different ionization modes and the differences between the analysis of free fatty acids and esterified fatty acids using this approach.

The authors should discuss the potential application (if any) of this approach for the determination of the position of double bonds of fatty acids in DAG and TAG species, a longstanding challenge in lipid analysis.

5. In figure 2, panel a, is elution time for the rest of phospholipid classes (if the bovine liver extract has more than PC, PE, PI and SM species).

6. In the last paragraph of page 21, it is indicated that separation was performed using a HILIC column, but the stationary phase (amine, amide, etc.) is not specified. Please do it.

7. In the last paragraph of page 22, it is stated that "PIS at m/z 153 was performed for profiling PGs and PIs in the negative ion mode". 153 is not specific for PI species, since it is the dehydrated ion of glycerol-phosphate, common to all phospholipid classes. It can be used for PA, but for PI the precursor ion scan of $m/z=241$ is more appropriate (dehydrated ion corresponding to inositol-phosphate). Please, consider changing this in the text.

8. In the caption of figure 1, the elution gradients description is confusing. Please clarify.

9. In the figure 3 it is stated that "comparison of positive mode ESI mass spectra of PC(16:0/18:1) after conducting the PB reaction in a flask for 10 minutes and through the flow microreactor for 40 s". Observing the panels c) and d) it is evident that the second reaction is cleaner. However, since data is shown as relative intensity (%), it is not possible to compare the intensity of the ions produced by both reaction. Please, as complementary information, include a Y-axis in the right part of the graphs with the absolute intensity of those peaks for both reactions.

Same for the supplementary figure 9, when it is stated that "positive ion mode mass spectra were used for comparison", but data is shown in relative intensity. Please include the Y-axis with absolute intensity in these graphs too.

10. In the supplementary figure 6, only the neutral loss of one fatty acid as ketene is shown in the central part of the spectra. Since neutral losses of fatty acids as ketene (and as carboxylate) are important diagnostic ions for the analysis of fatty acyl chains of phospholipids in negative mode, it is suggested that the authors to label the losses of the other fatty acid in the spectra as well.

Reviewer #3 (Remarks to the Author):

Analytically this is an impressive paper. One of the major challenges in lipidomics is sorting out where the double bonds are in these compounds. While exact mass can often annotate a formula and chromatography can separate by lipid class, fragmentation can only determine what fatty acids in terms of carbon number and number of double bonds contribute to the lipid species, and it has not been possible in most cases to determine where the double bonds are. This is an important deficit in our knowledge as diets have different contributions of specific fatty acids and the mammalian body handles these bonds differently, so understanding why the position of double bonds vary in intact lipids could have important impacts on our understanding of a wide range of lipid biochemistry. While others have suggested methods based on ozonolysis of double bonds and separation by ion mobility. For ozonolysis, applications have largely been limited to standards and a small number of lipids in extracts. For ion mobility the separation has been incomplete. Thus, a method whereby so many lipids can be characterised in relatively high throughput is likely to have a profound impact on the field.

While I have some specific comments my major criticism is the quality of the English. There are a number of sentences where the English is poor and while I understand the authors are probably not native speakers the manuscript would greatly benefit from a careful proof read. This is important as I think there is a lot of interesting work in this manuscript.

Specific questions and points:

An important component of this manuscript is the software they have generated. This needs to be made publically available, and I would suggest this is made available alongside a dataset so people can test the software and get the results from the paper. This could be made available through one of the Metabolomics repositories.

How easy is it to edit the lipid list in LOA to allow others to include different labelled standards for quantification such as deuterated and ¹³C labelled standards?

Figure 2. It might be worth explaining why the second numbers are larger than the first for the LC-MS/MS with and without PB. It's quite a modest number of lipids for an LC approach of an extract. Is this just the number of lipids that could be annotated or had reliable fragmentation data? This could be commented on.

Page 12. The authors switch nomenclature for fatty acids in a single sentence! They should stick to one nomenclature or the other: "the Δ 10 isomer (a very minor content) in C18:1, Δ 7(minor) and Δ 9 (major) isomer pair in C16:1, ω -6 (major) and ω -9 (minor) isomer pair in C18:2 and C20:2, and ω -3 and ω -6 isomer pair in C18:3, C20:3, C22:5..."

Page 12 "Notably, 48 new lipid species that are not listed in LIPID MAPS database, have been identified..." Is this true or is it that lipidmaps doesn't specify a position for the double bond?

Page 14. Have the Student t-test been corrected for multiple testing?

Page 15. "The above results are suggestive that the composition of free fatty acids may not reflect the composition of phospholipids containing the same acyl chains in tissue cells." I'm not sure what they mean by this sentence. Could they clarify this?

Page 18. What is the mechanistic basis for the discrimination of diabetic and non-diabetic samples? Is it be diet or genotype? This seems rather odd in that the discrimination is so large and the variation is so much smaller. Can they comment on this mechanism?

Page 20 "This aspect represents a huge advantage for clinical and point-of-care analysis, where small sample volumes are used and simple operation is highly appreciated." I think they might be overselling the approach at the moment.

References should be checked for consistency as some have full first names and some initials.

Supplementary Figure 15: Are two lipids really the only ones that have changed?

Supplementary figure 20: These haven't separated out the T2D samples and very few lipid change. Is this right?

Reviewers' comments:

Reviewer #1 (Remarks to the Author):

1. The relative quantitative of isomers, described on page 9, is based on product ion abundance ratios. To my knowledge this has not been independently validated for the intact glycerophospholipids, i.e., dissociation of [PC + 58 + H]⁺, [PE + 58 + H]⁺ etc. As presented the implication is that there is no PB reaction bias or dissociation effects associated with double bond position. This assumption is inconsistent with prior reports from this group (e.g., Ma et al. PNAS 2016, ref [22]) that show calibration curves derived from fatty acid standards [M-H]⁻ anions deviate significantly from a slope of 1. Further evidence needs to be provided therefore that these relative product ion abundances reflect relative isomer concentrations. If they do not, which I suspect is the case, this is not critical as the important result is visualization in the change in relative abundance NOT the relative abundance itself. Nonetheless the terminology used needs to clearly reflect the evidence.

Response: Agreed.

Changes: The terminology of “isomer (concentration) ratios” is change to “relative isomer ratios” throughout the manuscript. On page 9, the following sentence is added to further define “relative isomer ratios”.

“Because the standards of most lipid C=C location isomers are not commercially available, we could not perform isomeric quantitation. Nevertheless, abundance ratios of C=C diagnostic ions of the isomers do have linear correlations with isomer concentration ratios^{22,27}, which are used to indicate “relative isomer ratios” throughout the study”.

2. Pg 10 and elsewhere the term MW or molecular weight is used. MW has units of g/mol and is not relevant to the mono-isotopic measurements undertaken in a mass

spectrometer. Molecular mass or mass-to-charge ratio should be applied depending if the discussion pertains to neutrals or ions.

Changes: We use monoisotopic molecular mass with units of Dalton (Da) throughout the manuscript.

3. The analysis of both plasma, tissue and bovine liver lipidomes reports a number of odd chain species. E.g., pg. 11 PE 17:0_22:4. The authors should articulate how these are differentiated from isobaric ether-linked species (e.g., in this case PE(O-40:4)) or mono-methyl and dimethyl PE species, e.g., Me-PE38:4 (known to be present in liver – see Ejsing and co-workers *Biochimica et Biophysica Acta* 1811 (2011) 1081–1089). There may well be retention time information or negative ion CID product ions that supports the assignments made but this needs to be demonstrated/tabulated for cases where unusual or hitherto un-reported lipids are presented. This is important, as some of these samples have previously been heavily scrutinized and, for example, there is no consensus on odd chain contributions to the phospholipidome in plasma see (Quehenberger *Journal of Lipid Research* Volume 51, 2010 3299). More probably, in this instance at least the putative odd-chains are ether lipids.

Response: We are confident on the odd fatty acyl chain assignments because we obtained abundant odd fatty acyl anion signals from HPLC-MS² CID in negative ion mode. The list of fatty acyl fragment ions is provided in the Supporting Tables.

Isobaric methyl PE. The methyl PE would have produced fatty acyl anions and head group loss ions at quite different m/z in MS² CID and therefore would not interfere with the assignment.

Isobaric ether PE/PC. There could be co-existing ether PC or PE in those identified odd chain species. For instance, CID PE O-18:0_22:4 in negative ion mode should produce fatty acyl ions at m/z 331 for C20:4, overlapping C20:4 from isobaric PE 17:0_22:4. For the above situation, we choose not to report the ether structure because of lacking definitive evidence. In fact, we have confidently identified several ether lipids from the CID in negative ion mode with examples given in Supplementary Fig. 23 (PC

O-16:1_16:0). In those cases, their diacyl isobar is either at minor component or not present. More ideally, use of high-resolution mass analyzer should allow confident differentiation of the diacyl lipids from their isobaric ether lipids (0.03 Da difference). Discussion on ether lipids is added on page 10.

Change: A figure (Supplementary Fig. 11) is added as examples to illustrate identification of odd fatty acyl chains in PEs and PCs.

“Owing to fatty acyl chain analysis before C=C location determination, we also confidently identified a series of PCs and PEs GPs containing odd chains, e.g. C15, C17, C19 with 0-2 degrees of unsaturation (Supplementary Fig. 11). These data led us exclude some other possible isomeric structures, e.g. N-methyl PE²⁸. Ether GPs are isobaric to diacyl GPs but typically at lower abundances²⁹. In this study, we only report the ether lipids of which we have gathered confident MS/MS data (Supplementary Table 1). Given the limited mass resolving power from ion trap instrument, we could not identify ether GPs if they co-exist with diacyl GPs as a minor component and therefore they are not reported.”

4. Similar to (3) where unusual double bond locations are reported it should be clear how these have been assigned to particular chains. E.g., pg 13. A delta-7/delta-9 pair are assigned to 16:1 in a PC 14:0_16:1 but how is the possibility of a contribution from PC 14:1_16:0 excluded?

Changes: In order to clarify the process of C=C location determination, the following sentences are added on Page 9,

“It is worth noting that the fatty acyl chain composition needs to be determined first in the workflow of C=C location assignment. If there is only one or one dominant of unsaturated fatty acyl composition of the unknown lipid, which is the case for most GPs reported herein, the locations of C=C can be directly ascribed to that fatty acyl. For situations where multiple unsaturated fatty acyl compositions co-exist, the contribution of each isomer is estimated from the relative abundance of product ions from LC-MS/MS in negative ion mode. Then, the more abundant C=C diagnostic ions are

ascribed to the major species. For the minor components if the diagnostic ions have limited S/N or overlap with the more abundant species, we choose not to report the C=C location.”

5. Another example, is the putative delta-10 18:1 (18:1, n-8) on pg 12. I haven't found any mention of this fatty acid in the literature – what would the desaturase be? Are there literature (or other) GC data to support unassigned 18:1 species? It could be a branched chain Me-17:1(n-7) or alternatively the signal might arise from a contribution of a methylPE MPE(35:2). MPE and DMPE are known to be present in liver. Bilgin et al. *Biochimica et Biophysica Acta* 1811 (2011) 1081–1089

Response: The C18:1(Δ 10) could not be MPE as discussed in question 3 but could be a branched chain Me-17:1(n-7) fatty acyl, which is scarcely reported.

Changes: The following sentence is revised on page 13:

“Interestingly, for PE 18:1_18:1 and PC 18:1_18:1, we also detected relatively low intensity C=C diagnostic ions that could be assigned as the Δ 10 isomer, while other structural possibility may also exist (i.e. branched fatty acyl (Me-C17:1(Δ 9)).”

6. Pg 15. The double bond isomer ratios of phospholipids are compared to free fatty acids. Some more detail on the method for FFA determination should be provided. These methods are reasonably controversial and are subject to un-intended hydrolysis and background fatty acid contamination. For example, the solvents and glassware likely carry a significant abundance of oleic acid (it's always in my mass spectra!) this would likely skew the delta-7/delta-9 ratio reported unless there is careful examination of the background contribution in blank injections.

Response: FFAs were also analyzed by LC-PB-MS/MS method. The possibility of contamination of oleic acid from the environment was excluded from comparison to a blank injection, where a dominant product at m/z 183 was detected as the main interference (the blank injection spectrum was added in Supplementary Fig.19c).

Changes: Description on free fatty acid (FA 18:1) analysis is added into the section of Methods.

“For FA 18:1, the EPI mode in negative ion mode was applied for LC-PB-MS/MS analysis. The precursor ion was set as m/z 339.3, the collision energy was set as 45 eV for beam-type CID and 0.1 (arbitrary units) for ion trap CID.”

7. Isomer-level discrimination for Type-2 diabetes has been previously demonstrated by Stahlman et al. 2012 Diabetologia (DOI 10.1007/s00125-011-2444-6). This was only shown for a small set of isomers but it is nonetheless satisfying to see these differences in regulation of unsaturation mirrored across the lipidome.

Changes: The paper by Stahlman et al. is cited.

“...and C=C location isomers of certain TAGs have been found to change significantly in T2D plasma⁴¹”.

8. I could not locate Supplementary Figure 24 that is cited on pg 17.

Response: Supplementary figure numbers are corrected.

9. I could not locate key supplementary Tables.

Response: Supplementary Tables are added.

10. Pg 17. Figure caption states “Precision of isomer ratio measurements is much less affected by subject variations as compared to relative quantitation measurements.” This is a very clear articulation of a critical point. To some degree it is lost in the figure caption and deserves a bit more discussion. This finding suggests tighter

regulation/dysregulation at the isomer level which is unaffected by dilution or other normalization complexity (discussed above). It is also consistent with recent isomer-resolved imaging MS studies which have highlighted tissue-tissue and tissue-tumour differences at the isomer level that are invisible at the molecular lipid level. <https://doi.org/10.1002/anie.201806635> and <https://doi.org/10.1002/anie.201802937>

Response: The following discussions are added on page 20 and the references are cited. “The finding of high precision in measuring relative ratio of lipid C=C location isomers suggests tighter regulation/dysregulation at the isomer level which is unaffected by sample complexity or preparation procedures. These results are also consistent with recent isomer-resolved imaging MS studies which have highlighted tissue-tissue and tissue-tumor differences at the isomer level that are invisible at the molecular lipid level 44,45”

11. Finally, the demonstrated efficacy of double bond positional isomers as biomarkers shown here sets a clear challenge for the mass spectral discrimination of other types of lipid regio- and stereo-isomers.

Response: agreed.

12. I found some of the English language in the introduction a bit hard to read. I have suggested a few edits as annotations in the attached PDF.

Response: We highly appreciate the edits and have improved the English according to the reviewer’s suggestions.

Reviewer #2 (Remarks to the Author):

This is a generally well-written manuscript in which the authors, continuing their previous work, demonstrate that their analytical platform can provide comprehensive and detailed information about the location of double bonds in the fatty acyl chains of PC and PE phospholipids. The analytical approach expands the classic workflow for the determination of phospholipid species (including a liquid chromatography step to separate phospholipid classes, a positive ion analysis to confirm the polar head of phospholipid and a subsequent negative analysis to characterize the fatty acid composition of the molecules) with the inclusion of an online derivatization step of the fatty acids double bonds based on the Paternò-Büchi reaction. This derivatization provides unique information about the position of the double bonds in the fatty acyl chains of phospholipids when they are fragmented in positive ion mode. The inclusion of the derivatizing agent acetone in the mobile phase and the post-column derivatization is both clever and convenient. It is also worth noting that an in-house software, Lipid Omega Analyzer, was developed to help with the analysis of data. As proof-of-concept studies, Zhang et al., use the platform to characterize the phospholipidome of human breast cancer tissue and plasma from patients suffering type 2 diabetes with significant success.

Even though the Paternò-Büchi reaction for the characterization of fatty acid positional isomers has been introduced before by this group and others, although its online incorporation as part of the analytical methodology represents a significant accomplishment, yielding results never achieved before for the comprehensive analysis of individual phospholipid species. Although the paper provides a well-designed approach for the full characterization of some phospholipid species, some examples of the analysis of other major phospholipid classes in mammals (PI and PS, especially), as well as other minor phospholipid species that play a role in human physiology (PA, PG, BMP and CL) would be useful.

All in all, there are no major concerns that preclude the publication of this article, however, some minor issues should be correctly address before.

Minor concerns:

1. The title does not take advantage of the content of the manuscript. Especially the fact of describing this approach as suitable for “biomarker discovery”. It is true that the ratios between different positional isomers can provide a new layer of information for the discovery of biomarkers. However, pointing it out in the title may be misunderstood by readers as a specific approach for biomarker discovery and thus, reduce the potential of this approach for other biomedical applications that are nowadays at the forefront of lipid science. For instance, positional isomers of palmitoleic acid, sequentially measured using time-consuming GC/MS approaches have shown different effect modulating the inflammatory response (doi: 10.1194/jlr.M079145). With this approach, all 3 isomers could have been identified, quantified and tested for biological effect as part of the same study, considerably reducing time and costs. Other example is the measurement of omega-3 and omega-6 docosapentaenoic acid isomers (known to have different biological effects) esterified in phospholipids at the same time.

The authors are encouraged to consider changing the title of the manuscript to other either more general (“Comprehensive Analysis of Unsaturated Phospholipid Isomers for New Biomedical Applications” or similar) or more focused in the analytical approach (“Comprehensive Analysis of Fatty Acids Positional Isomers of Phospholipids Using an Online Photochemical Derivatization” or similar). A change in the title could make the paper more attractive for readers and increase its scope. As a side comment, the analysis is constrained to PC and PE species, the inclusion of the term “Phospholipids” does not fit with the content of the manuscript.

Response: We appreciate the suggestions by the reviewer and we love the title “Comprehensive Analysis of Unsaturated Phospholipid Isomers for New Biomedical Applications”.

Changes: Discussion about identification of C16:1 isomers is added into the section of “Analysis of GP from bovine liver polar extract”.

“For fatty acyl C16:1, $\Delta 7$ (minor, has been reported previously by GC-MS³⁰) and $\Delta 9$ (major) isomer pair was detected.”

2. Even though the analysis of other phospholipid classes (PI, PS, PA, PG, BMP and CL) is sometimes mentioned in the text, MS/MS spectra for these lipid classes are not shown. Authors are requested to show at least one MS/MS spectrum of the identification of positional isomers of fatty acids of some of these phospholipid classes. Since the analysis is carried out in positive mode, at least one example of PS and PG molecules should be found. While the ionization in positive mode for PI, PA and CL is poor, the authors are encouraged to find one example of these molecules (for instance the major PI species PI(18:0/20:4)). The inclusion of more examples could generalize the model for the analysis of the entire phospholipidome.

Response: Identification of C=C positions of PI and PG are achieved in negative ion mode (MS³), which is more sensitive for these lipids. A supplementary figure is added to illustrate PG and PI analysis. The LC-PB-MS/MS platform needs to be tailored for lipid molecules of lower abundance in future studies.

Changes: Mass spectra of C=C identification of PG 34:1, PI 36:1 and PI 38:4 in bovine liver extracts were added in Supplementary Fig.10.

3. Related with the previous concern, ether-linked fatty acids of PC and PE species are not shown in the analysis. This is especially relevant for PC and PE species, where alkyl or vinyl-ether linked fatty acids represent an important part of these phospholipid pools (in PE can represent up to 50% of total mass) and whose dysregulation is related to several diseases (<https://doi.org/10.1016/j.bbadis.2012.05.014>). Is there any difference in the fragmentation behavior of ether-linked fatty acids compared to their ester-linked counterparts for the identification of double bonds position. Please, show a comparative MS/MS spectra of PC or PE species containing the same unsaturated fatty acid in the sn-1 position, but linked with an ester or an ether bond.

Response: We identified limited number of ether-linked fatty acids of PC and PE, likely due to their low relative concentrations (typically in the range of <1% of most abundance species). However, these identifications are highly confident. Some examples are shown in supplementary Figure 21, supplementary Table 1, 2, and 3.

Changes: Discussion about ether-linked fatty acids of phospholipids is added on page 10, an example is shown in Supplementary Fig. 23. The suggested reference is cited.

“Ether GPs are isobaric to diacyl GPs but typically at lower abundances²⁹. In this study, we only report the ether lipids of which we have gathered confident MS/MS data (Supplementary Table 1). Given the limited mass resolving power from ion trap instrument, we could not confidently identify ether GPs if they co-exist with diacyl GPs as a minor component and therefore they are not reported.”

4. A recent paper describing the Paternò-Büchi reaction to determine the position of double bonds in free fatty acids has been published (10.1021/acs.analchem.7b02375). Since this paper uses negative ion mode to characterize the fatty acids and discuss about mechanistic aspects of the fragmentation, the authors should cite it, as well as, to discuss briefly the use of different ionization modes and the differences between the analysis of free fatty acids and esterified fatty acids using this approach. The authors should discuss the potential application (if any) of this approach for the determination of the position of double bonds of fatty acids in DAG and TAG species, a longstanding challenge in lipid analysis.

Changes: The suggested work is now cited (Reference 23). Discussion about analysis of fatty acids and other phospholipids, and potential applicability of LC-PB-MS/MS system is added.

“LC-PB-MS/MS was successful in analyzing unsaturated free fatty acids, PG, and PI in negative ion mode. Although the PB products of unsaturated glycerolipids were detected ($[\text{P}^{\text{B}}\text{M}+\text{NH}_4]^+$ or $[\text{P}^{\text{B}}\text{M}+\text{Na}]^+$), separation and ionization conditions needed to be tailored for these neutral lipids from biological samples.”

5. In figure 2, panel a, is elution time for the rest of phospholipid classes (if the bovine liver extract has more than PC, PE, PI and SM species).

Changes: Elution time of 20 min is adequate to elute most phospholipids species.

“This gradient elution is adequate for separating most phospholipids in biological samples, including PE, PC, PG, PA, PI, PS and SM²⁵”.

6. In the last paragraph of page 21, it is indicated that separation was performed using a HILIC column, but the stationary phase (amine, amide, etc.) is not specified. Please do it.

Response: The stationary phase is silica spheres; this information is added into the section of “LC-PB-MS platform” in the Methods.

7. In the last paragraph of page 22, it is stated that “PIS at m/z 153 was performed for profiling PGs and PIs in the negative ion mode”. 153 is not specific for PI species, since it is the dehydrated ion of glycerol-phosphate, common to all phospholipid classes. It can be used for PA, but for PI the precursor ion scan of $m/z=241$ is more appropriate (dehydrated ion corresponding to inositol-phosphate). Please, consider changing this in the text.

Changes: Revision made accordingly:

“PIS at m/z 153 was performed for profiling PGs and PIS 241 was performed for profiling PIs in the negative ion mode”.

8. In the caption of figure 1, the elution gradients description is confusing. Please clarify.

Changes: The elution gradients description is revised.

“A started from 90%, decreased to 85% at 5 min, then decreased to 80% at 8 min, kept at 80% within 8-15 min, and decreased to 70% in 16 min and kept this percentage to 20 min”.

9. In the figure 3 it is stated that “comparison of positive mode ESI mass spectra of PC (16:0/18:1) after conducting the PB reaction in a flask for 10 minutes and through the flow microreactor for 40 s”. Observing the panels c) and d) it is evident that the second

reaction is cleaner. However, since data is shown as relative intensity (%), it is not possible to compare the intensity of the ions produced by both reaction. Please, as complementary information, include a Y-axis in the right part of the graphs with the absolute intensity of those peaks for both reactions.

Changes: Peak intensities are added in Supplementary Fig. 3c-d.

Same for the supplementary figure 9, when it is stated that “positive ion mode mass spectra were used for comparison”, but data is shown in relative intensity. Please include the Y-axis with absolute intensity in these graphs too.

Changes: Peak intensities are added in the mass spectra (Supplementary Fig. 12) for comparison of nanoESI and LC-MS.

10. In the supplementary figure 6, only the neutral loss of one fatty acid as ketene is shown in the central part of the spectra. Since neutral losses of fatty acids as ketene (and as carboxylate) are important diagnostic ions for the analysis of fatty acyl chains of phospholipids in negative mode, it is suggested that the authors to label the losses of the other fatty acid in the spectra as well.

Changes: The losses of the fatty acid chains are added in the supplementary Fig. 6.

Reviewer #3 (Remarks to the Author):

Analytically this is an impressive paper. One of the major challenges in lipidomics is sorting out where the double bonds are in these compounds. While exact mass can often annotate a formula and chromatography can separate by lipid class, fragmentation can only determine what fatty acids in terms of carbon number and number of double bonds contribute to the lipid species, and it has not been possible in most cases to determine where the double bonds are. This is an important deficit in our knowledge as diets have different contributions of specific fatty acids and the mammalian body handles these bonds differently, so understanding why the position of double bonds vary in intact lipids could have important impacts on our understanding of a wide range of lipid biochemistry. While others have suggested methods based on ozonolysis of double bonds and separation by ion mobility. For ozonolysis, applications have largely been limited to standards and a small number of lipids in extracts. For ion mobility the separation has been incomplete. Thus, a method whereby so many lipids can be characterized in relatively high throughput is likely to have a profound impact on the field.

While I have some specific comments my major criticism is the quality of the English. There are a number of sentences where the English is poor and while I understand the authors are probably not native speakers the manuscript would greatly benefit from a careful proof read. This is important as I think there is a lot of interesting work in this manuscript.

Specific questions and points:

1. An important component of this manuscript is the software they have generated. This needs to be made publically available, and I would suggest this is made available alongside a dataset so people can test the software and get the results from the paper. This could be made available through one of the Metabolomics repositories.

Response: we currently are working hard on making the software available on a cloud server so other researchers could upload their own data collected using the PB method

and get them analyzed. The current version and the data set provided for review will be made available upon request.

2. How easy is it to edit the lipid list in LOA to allow others to include different labelled standards for quantification such as deuterated and ¹³C labelled standards?

Response: there is no fundamental technical barrier for realizing this. All it needs is to include the list of the internal standards and the established calibration curves (equations) in the data base. The following discussion is added in the Methods/Data Analysis:

“With proper information added in the database for internal standards and corresponding calibration curves, the capability of LOA can be extended for absolute quantitation.”

3. Figure 2. It might be worth explaining why the second numbers are larger than the first for the LC-MS/MS with and without PB. It's quite a modest number of lipids for an LC approach of an extract. Is this just the number of lipids that could be annotated or had reliable fragmentation data? This could be commented on.

Response: Due to the nature of sample (supplied as the polar extract from bovine liver), the identified species are limited to GPs which makes the ID number seemly modest. Because a significant portion of GPs containing C=C location isomers, the molecular species identified by LC-PB-MS/MS (Fig. 2h) are larger than lipids identified at fatty acyl level (Fig. 2g). All these lipids are confidently assigned by MS/MS.

Changes: The following sentence is added to clarify Fig. 2g and 2h.

“Because a significant portion of GPs containing C=C location isomers, the molecular species identified by LC-PB-MS/MS are more than the lipids identified at fatty acyl level.”

4. Page 12. The authors switch nomenclature for fatty acids in a single sentence! They should stick to one nomenclature or the other: “the Δ 10 isomer (a very minor content) in C18:1, Δ 7(minor) and Δ 9 (major) isomer pair in C16:1, ω -6 (major) and ω -9 (minor) isomer pair in C18:2 and C20:2, and ω -3 and ω -6 isomer pair in C18:3, C20:3, C22:5...”

Changes: Thanks for the comments. The sentences are revised as follows:

“Moreover, C=C location isomers of poly unsaturated fatty acyl (PUFA) chains from a series of PC and PE were also identified using LC-PB-MS/MS. For simplicity, the omega (ω) nomenclature which commonly referred for PUFA is used here; these lipid isomers include ω -6 (major) and ω -9 (minor) isomer pair in C18:2 and C20:2, and ω -3 and ω -6 isomer pair in C18:3, C20:3, C22:5 (Supplementary Fig. 9, Supplementary Table 1)”.

5. Page 12 “Notably, 48 new lipid species that are not listed in LIPID MAPS database, have been identified...” Is this true or is it that lipidmaps doesn't specify a position for the double bond?

Changes: We clarify the sentence as: “Notably, 48 new lipid species that are not listed in LIPID MAPS database for C=C locations, have been identified (highlighted in Supplementary Table 1).”

6. Page 14. Have the Student t-test been corrected for multiple testing?

Response: We did not perform multiple testing before. In the revision, discussion about correction of the t-test is added on page 16.

Changes: “To correct false discovery, multiple testing was performed³⁶. Significant changes were still observed for these lipid isomers, except for PC 19:0_18:1 (corrected value of -0.0005)”.

7. Page 15. “The above results are suggestive that the composition of free fatty acids may not reflect the composition of phospholipids containing the same acyl chains in tissue cells.” I’m not sure what they mean by this sentence. Could they clarify this?

Changes: This sentence is revised as:

“The above results suggest that although the free fatty acids and the corresponding fatty acyls in phospholipids may contain the same pair of C=C location isomers, their ratios can be quite different or even unrelated.”

8. Page 18. What is the mechanistic basis for the discrimination of diabetic and non-diabetic samples? Is it be diet or genotype? This seems rather odd in that the discrimination is so large and the variation is so much smaller. Can they comment on this mechanism?

Response: These are very good questions! We do not have answers and could not find relevant references. Thus, we are reluctant to comment or propose any mechanism for the observed phenomenon.

9. Page 20. “This aspect represents a huge advantage for clinical and point-of-care analysis, where small sample volumes are used and simple operation is highly appreciated.” I think they might be overselling the approach at the moment.

Response: We believe this statement is generally true for any quantitative MS methods if internal standards are not needed. A main challenge for transferring mass spectrometry analysis for point-of-care application is the complex procedure that cannot be allowed to perform onsite in clinics. Incorporation of internal standards is particular troublesome since accurate control of the sample/solvent volumes as well as use of relatively large amounts of samples are required to minimize the errors in quantitation.

10. References should be checked for consistency as some have full first names and some initials.

Changes: Thanks for pointing out the mistakes. The format of references is corrected.

11. Supplementary Figure 15: Are two lipids really the only ones that have changed?

Response: The heatmap is mainly for visual comparison and clustering. Supplementary Figure 13 has better comparisons for significant changes.

12. Supplementary figure 20: These haven't separated out the T2D samples and very few lipid change. Is this right?

Response: Correct! No significant changes were detected from the cohort from the subclass level analysis.

REVIEWERS' COMMENTS:

Reviewer #1 (Remarks to the Author):

The authors have carefully responded to all the points raised in my initial review and significantly improved the manuscript. As it stands this is a very important contribution to the field of lipidomics and should be published.

Reviewer #2 (Remarks to the Author):

The authors have addressed all of the concerns raised and the manuscript is suitable for publication.

Reviewer #3 (Remarks to the Author):

I am happy with all the changes that the authors have made and thank them for addressing my comments. The only concern I have is that they are still in the process of making their software available. I feel this will be an important tool to test the validity of their approach. Can this software and a dataset not be hosted by the journal to allow readers of the paper to replicate the approach described? I think this is an important point and consistent with the journal's open access policy for data and tools.

Reponses to reviewers' comments:

Reviewer #1 (Remarks to the Author):

Th authors have carefully responded to all the points raised in my initial review and significantly improved the manuscript. As it stands this is a very important contribution to the field of lipidomics and should be published.

Reviewer #2 (Remarks to the Author):

The authors have addressed all of the concerns raised and the manuscript is suitable for publication.

Reviewer #3 (Remarks to the Author):

I am happy with all the changes that the authors have made and thank them for addressing my comments. The only concern I have is that they are still in the process of making their software available. I feel this will be an important tool to test the validity of their approach. Can this software and a dataset not be hosted by the journal to allow readers of the paper to replicate the approach described? I think this is an important point and consistent with the journal's open access policy for data and tools.

Response: We have uploaded the LOA software and raw data onto GitHub. They are now publicly accessible at <https://github.com/LipidAnalysis?tab=repositories>.